# Oral Ingestion of Yuzu Seed Oil Suppresses the Development of Atopic Dermatitis-like Skin Lesions in NC/Nga Mice

**DOI:** 10.3390/ijms25052689

**Published:** 2024-02-26

**Authors:** Kimito Asano, Yoshiya Watanabe, Mio Miyamoto, Mochifumi Toutani, Shunji Mizobuchi

**Affiliations:** 1Kochi-Umajimura Yuzu Health Research Course, Kochi Medical School, Kochi University, Nankoku 783-8505, Japan; 2Umajimura Agricultural Cooperative, Kochi 781-6201, Japan

**Keywords:** yuzu seed oil, limonin, atopic dermatitis, Nrf2, antioxidant effect

## Abstract

Long-term oral ingestion of unheated yuzu seed oil in humans reduces lipid peroxides in the blood. Moreover, yuzu seed oil contains limonin, which can induce antioxidant and anti-inflammatory effects by activating the transcription factor nuclear factor erythroid 2-related factor 2 (Nrf2). Previously, Nrf2 has been shown to reduce atopic dermatitis (AD). Therefore, we hypothesized that ingesting unheated yuzu seed oil can regulate AD through Nrf2. An AD model was established using NC/Nga mice through repeated local exposure to mite antigens. Unheated and purified yuzu seed oil (100 µL/mice) or water (control, 100 µL/mice) was administered orally once a day using a gastric cannula for rodents for 28 days. On day 28, mice in the unheated yuzu seed oil group exhibited significantly lower clinical skin severity scores and ear thickness than those in the purified yuzu seed oil and water groups. Serum histamine levels remained unaltered among the three AD-induced groups. Serum *Dermatophagoides farina* body (Dfb)-specific immunoglobulin E (IgE) levels were significantly lower in the unheated yuzu seed oil group. Oral ingestion of yuzu seed oil in NC/Nga AD model mice significantly suppressed dermatitis deterioration and decreased serum IgE levels. Clinical trials (*n* = 41) have already confirmed that unheated yuzu oil is safe for long-term intake, further suggesting its potential use in improving AD symptoms.

## 1. Introduction

Yuzu (*Citrus junos* Sieb. ex Tanaka) is a typical Japanese citrus fruit with a desirable aroma. Yuzu juice and peel, which are originally from China and arrived in Japan via Korea 1000 years ago, are characterized by a sour taste and refreshing aroma and are used as raw materials for vinegar and seasonings in Japan. Although yuzu seeds constitute up to 9% of the fruit weight, they are typically discarded after juice extraction because of their perceived lack of utility. However, recognizing the richness of active ingredients in the seeds, we extracted unheated yuzu seed oil from yuzu seeds using the cold-press method, subsequently exploring its effect. Component analysis of citrus seed oil showed that it contains a large amount of limonoids (e.g., limonin, nomilin, and obacunone) [1,2,3]. Therefore, research is being conducted on the effects of limonoids contained in yuzu seed oil with respect to their anti-oxidant activity [4,5,6].

We have previously demonstrated that long-term oral intake of unheated yuzu seed oil in humans effectively reduced lipid peroxides in the blood [7], showcasing its antioxidant effect. The transcription factor nuclear factor erythroid 2-related factor 2 (Nrf2), which plays an important role in mitigating lipid peroxidation, is a key regulator of cellular antioxidant responses and controls the expression of genes that counteract oxidative and electrophilic stresses [8,9]. In the nucleus, Nrf2 binds to antioxidant response element (ARE) sequences along with the small musculoaponeurotic fibrosarcoma oncogene homologue (sMaf) to regulate the expression of more than 200 antioxidant and anti-inflammatory genes [10,11,12]. Therefore, it has been suggested that unheated yuzu seed oil contains certain components that activate Nrf2 and suppress reactive oxygen species (ROS) production.

Atopic dermatitis (AD) is a chronic inflammatory skin disease disorder characterized by eczema and repeatedly exhibits exacerbation and remission of recurrent eczematous lesions [13]. AD pathology comprises three factors: (1) skin barrier dysfunction, (2) immune dysregulation (type-2 helper T (Th2) dominance or interleukin (IL)-4/IL-13 axis inflammation), and (3) intense itching, which occur independently or mutually [14,15,16]. IL-4 and IL-13 activate dual oxidase protein 1 (DUOX1) and generate ROS [17]. Oxidative stress is one of the most important pathogenic factors of AD [18]. Patients with AD are more susceptible to damage caused by ROS or oxidizers than controls, which is evident from an increase in malondialdehyde and a decrease in enzymatic and non-enzymatic antioxidants [19]. Oxidative stress and changes in antioxidant defenses are involved in the pathophysiology of acute exacerbations of AD [20]. These results suggest that antioxidant supplements may be beneficial in treating AD [18].

Studies using AD mouse models and analyses of human AD samples have revealed that Nrf2 improves AD inflammation [21,22,23,24,25,26]. Activating Nrf2 by oral intake of a natural or chemical substance improved inflammatory signaling in human keratinocytes in vitro as well as sensitizer-induced skin inflammation in a mouse model of human AD.

In this study, we evaluated the effects of consuming yuzu seed oil in NC/Nga mice, an established animal model for human AD.

## 2. Results

### 2.1. Effects of Yuzu Seed Oil on Inflammation Score and Ear Thickness

The non-sensitized group, in which the mite antigen was not applied, showed no development of human AD-like skin lesions, resulting in a clinical skin severity score of zero.

Conversely, in the three AD-induced groups, topical application of the antigen immediately induced clinical symptoms, such as erythema, edema, and hemorrhage on the ears and back. This was followed by superficial erosion, deep excoriation, scarring, and skin dryness, all of which were markedly reduced in both unheated and purified yuzu seed oil groups (Figure 1). On day 28, the clinical skin severity score of the water, unheated, and purified yuzu oil groups were 4.0 ± 0.71 (*n* = 5), 0.4 ± 0.89 (*n* = 5; *p* < 0.001 vs. water group), and 2.0 ± 1.00 (*n* = 5; *p* = 0.0064 vs. water group) points, respectively. A significant difference was observed between the unheated and purified yuzu seed oil group (*p* = 0.028; Figure 2).

The ear thicknesses in the different groups were as follows: 0.27 ± 0.019, 0.41 ± 0.073, 0.28 ± 0.017 (*p* < 0.001 vs. water group), and 0.37 ± 0.057 mm (each *n* = 5), for non-sensitized, water, and unheated and purified yuzu seed oil groups, respectively. A significant difference was also observed between the unheated and purified yuzu seed oil groups (*p* < 0.001). In contrast, no differences were observed between the purified yuzu seed oil and water groups (Figure 3).

### 2.2. Yuzu Seed Oil Did Not Inhibit an Increase in Serum Histamine Levels

The serum histamine level (in nM) of the non-sensitization group was 0.02 ± 0.007 (*n* = 5). Conversely, the histamine level of the water group, which was sensitized with the antigen under specific pathogen-free (SPF) conditions, increased to 0.97 ± 0.360 (*p* < 0.001). Among the three sensitized groups, the unheated yuzu seed oil group showed lower values (0.66 ± 0.212), but no significant differences were observed among the three groups (Figure 4).

### 2.3. Inhibitory Effect of Yuzu Seed Oil on Serum Dfb-Specific IgE Levels

The serum *Dermatophagoides farina* body (Dfb)-specific IgE levels demonstrated a significant correlation with dermatitis progression. The oral administration of yuzu seed oil to NC/Nga mice reduced the elevation of serum Dfb-specific IgE levels. The level of IgE was represented by the absorbance at A_450_. The serum Dfb-specific IgE level of the water group was 1.05 ± 0.442 (*n* = 5), which was significantly higher than that of the non-sensitized group, 0.14 ± 0.035 (*n* = 5; *p* < 0.001). Furthermore, the serum Dfb-specific IgE levels of the unheated and purified yuzu seed oil groups were 0.34 ± 0.136 (*n* = 5; *p* = 0.0018 vs. water group) and 0.51 ± 0.162 (*n* = 5; *p* = 0.011 vs. water group), respectively. Moreover, unheated yuzu seed oil was significantly more effective than purified yuzu seed oil (*p* = 0.048; Figure 5).

## 3. Discussion

Limonin, a limonoid belonging to the triterpenoid group, is a bitter component of citrus seeds and fruit tissues. The unheated yuzu seed oil used in this study contained limonin, which has been shown to regulate the expression of various cytoprotective enzymes such as heme oxygenase-1 (HO-1) and NAD(P)H quinone oxidoreductase 1 (NQO1) by activating the Nrf2/ARE pathway [27,28,29]. Moreover, inducing the Nrf2/HO-1 signaling pathway inhibits inflammatory AD in skin keratinocytes [30]. Therefore, we hypothesized that limonin-containing unheated yuzu seed oil plays a regulatory role in AD via the Nrf2/ARE pathway.

In the present study, the clinical skin severity score and ear thickness of the water group (negative control), which was sensitized with the antigen under SPF conditions, increased from sensitization onset. In contrast, oral ingestion of unheated yuzu seed oil significantly suppressed the visual appearance of dermatitis and increased the clinical skin severity score, ear thickness, and serum Dfb-specific IgE levels compared with the water and purified yuzu seed oil groups. Yamamoto et al. [31] have investigated the role of scratching behavior in a Dfb-induced dermatitis model in NC/Nga mice. They revealed that the severity of dermatitis correlated with increasing scratching behavior. Itch is an unpleasant sensation that may prompt desire to scratch the affected area [32]. Therefore, an increase in scratching behavior is accompanied by intense itching. Therefore, the results of this study indicate that oral intake of unheated yuzu seed oil reduces itching, which is one of the etiological factors of AD.

Regarding itch, there are at least two types of afferents, histamine-dependent and histamine-independent types, which indicates that the effect of antihistamines in treating AD is limited [33,34]. In this study, the unheated yuzu seed oil group had good skin scores and auricle thickness; however, no significant difference was observed in blood histamine levels compared with the control and purified yuzu seed oil groups. These findings suggest that unheated yuzu seed oil suppresses non-histaminergic pathways in atopic itching.

Th2 cells release IL-4 and IL-13, leading to IgE class switching in B cells and production of antigen-specific IgE via the signal transducer and activator of transcription (STAT) pathway [13,35]. In this study, serum IgE production was suppressed in the unheated yuzu seed oil group compared to that in the control and purified yuzu seed oil groups. Th2 cytokine production is negatively regulated by Nrf2 [36], suggesting that limonin in unheated yuzu seed oil activates Nrf2, thereby suppressing IL-4 and IL-13. Moreover, IL-4 and IL-13 have been implicated in chronic itch [37]. Histamine, which has been reported to cause itching, did not differ among the three groups in this study. Limonin-activated Nrf2 suppressed IL-4 and IL-13 production, potentially suppressing itching.

Moreover, dorsal skin symptoms that could not be scratched were significantly alleviated in the unheated yuzu seed oil group. ROS play an important role in AD development. Oxidative stress may be an intrinsic mediator of amplification and chronicity in AD [38]. Limonin activates Nrf2 and exerts antioxidant effects. Qin et al. [39] have shown that limonin extracted from pomelo seeds dose-dependently enhanced the transcription of Nrf2 and its downstream genes HO-1 and NQO1 in HepG2 cells, and also stimulated the expression of Nrf2, HO-1, and NQO1. Based on the above results, we conclude that limonin exerts its antioxidant properties by activating the Nrf2/ARE pathway. Moreover, Li et al. [40] have established a model of NAFLD in zebrafish larvae and shown that limonin treatment induced antioxidant effects by upregulating the Nrf2/HO-1 signaling pathway in the liver, showing a protective effect against NAFLD. We have previously demonstrated that long-term oral intake of unheated yuzu seed oil in humans reduces lipid peroxides in the blood [7]. Therefore, it was further confirmed that unheated yuzu seed oil has an antioxidant effect. This antioxidant effect was attributed to Nrf2 activation by the limonin contained in unheated yuzu seed oil, and this antioxidant effect improved the skin lesions on the back.

Activation of the Nrf2-ARE pathway has been reported to improve the symptoms of dermatitis in an AD model. Choi et al. [21] have shown that saponins derived from *Platycodon grandiflorus* root attenuated AD-like skin lesions by activating Nrf1/ARE-mediated OH-1 in a 2,4-dinitrochlorobenzene (DNCB)-induced AD mouse model. Park et al. [23] have reported that 6-Shogalo, an active compound in ginger, alleviates DNCB-induced AD-like skin lesions and scratching behavior by inhibiting immune mediators, including IL-4 and IL-13, via regulation of the ROS/mitogen-activated protein kinase (MAPKs)/Nrf2 signaling pathway. Karuppagounder et al. [22] have demonstrated that quercetin treatment in NC/Nga mice attenuated *D. farinae* extract-induced AD-like clinical symptoms via the induction of Nrf2 signaling pathways. As a result, quercetin inhibited IL-4, an inflammatory cytokine. Choi et al. [26] have reported that oral administration of miquelianin, which activates Nrf2 ex vivo, suppresses ear thickening and inflammatory cell infiltration, reduces serum IgE production in trimellitic anhydride-induced AD model mice, and alleviates IL-13 production in lymph nodes. As described above, Nrf2 stimulation suppresses type 2 immunity or IL-4/IL-13 axis inflammation, a significant etiological factor in AD.

Although the purified yuzu seed oil used in this study contained limonin below the detection limit, the skin severity scores and Dfb-specific serum IgE levels were significantly lower than those of the control (water group). This suggests that yuzu seed oil may contain active ingredients other than limonin that are not lost during volatile purification. However, additional studies are required to confirm their presence.

## 4. Materials and Methods

### 4.1. Oils Used for Intervention

The yuzu seed oil used in this study was extracted from yuzu seeds using a type AP 10 Reinartz screw press (Maschinenfabrik Reinartz, Neuss, Germany), with the operating frequency set to 62 Hz for the cold-pressing of oil seeds. The extract was filtered and used as unheated yuzu seed oil. Unheated yuzu seed oil was purified by using vaporization to generate purified yuzu seed oil. Both oils were bottled without any additives and stored in unlabeled 200 mL brown glass bottles at +5 °C until use.

The unheated yuzu seed oil exhibited a fatty acid composition of 36.8% oleic acid, 35.6% linoleic acid, and 20.1% palmitic acid. The purified yuzu seed oil contained 37.0% oleic acid, 35.4% linoleic acid, and 19.9% palmitic acid (measured by using gas chromatography, Japan Food Research Laboratories, Tokyo, Japan). Furthermore, an X-LC system (JASCO Corporation, Tokyo, Japan) at the Kochi Prefectural Industrial Technology Center was used to reveal that 100 mL of unheated yuzu seed oil contained 49.4 mg or more of nomilin, 10.4 mg or more of limonin, and 1.88 mg or more of obakunon. However, the purified yuzu seed oil had nomilin, limonin, and obakunon contents below the detection limit.

### 4.2. Animals and Ethics

For the experiments, 10-week-old NC/Nga mice were used as the mouse model of AD [41]. NC/Nga mice developed human AD-like skin lesions with elevated serum IgE levels when bred under conventional conditions [42]. They were purchased from Charles River Laboratories Japan (Yokohama, Japan).

The animals were maintained in HEPA-filtered air at our facility. All animals were healthy; five mice were housed per cage in an air-conditioned animal room at 23 ± 2 °C, a relative humidity of 50 ± 5%, in light and dark cycles of 12 h each, and fed with a laboratory diet and water ad libitum.

All animal experiments were approved by the Animal Study Committee of Kochi Medical School according to the government guidelines for animal care (protocol code: E-00010).

### 4.3. Sensitization and Development of AD Mouse Model

In vivo experiments on AD-like skin lesions were induced by repeated topical application of Dfb extract [31] (Biostir AD, Biostir Inc. 20, Kobe, Japan). After complete shaving of the dorsal hair and depilation using a hair removal cream, the backs and ears of the mice were sensitized by applying 100 mg Biostir AD (Biostir Inc. 20), except for the non-sensitized group, three times a week for 2 weeks following the first challenge (Figure 6).

### 4.4. Oral Administration Method of Yuzu Seed Oil

We divided NC/Nga mice into four groups: a normal healthy group (non-sensitized group, *n* = 5) and three AD-induced groups followed by orally administering water, yuzu juice, or yuzu seed oil (100 µL/mice by an oral gastric cannula for rodents once a day for 28 days); a negative control (water group, *n* = 5), unheated yuzu seed oil (*n* = 5), and purified yuzu seed oil groups (*n* = 5).

### 4.5. Clinical Estimation of Skin Lesions in NC/Nga Mice

The backs and ears of the NC/Nga mice were clinically observed for 4 weeks. The clinical severity of dermatitis was recorded using macroscopic diagnostic criteria. Briefly, dermatitis severity was evaluated on day 28 by assessing four specific criteria: the severity of dermatitis was assessed macroscopically by an AD-scoring method [41], in which the degree of each symptom (erythema, edema/thickness, hemorrhage/excoriation, and dryness) was scored as 0 (absent), 1 (mild), 2 (moderate), or 3 (severe). The sum of individual scores was used as the dermatitis score.

### 4.6. Measurement of Ear Thickness

On day 28, the thicknesses of the left and right ears were measured five times using a micrometer equipped with a constant-pressure device (MDQ-30, Mitsutoyo Corporation, Kanagawa, Japan) under light ether anesthesia. The average thickness of both ears was calculated and considered as ear swelling.

### 4.7. Measurement of Histamine in the Serum

Serum histamine levels were measured using ELISA (Histamine enzyme immunosorbent kit). The kit was purchased from BIO connect (Montigny Le Bretonneux, France).

### 4.8. Measurement of Serum Dfb-Specific IgE Levels

The serum levels of Dfb-specific immunoglobulin E (IgE) were examined on day 28. For the Dfb-specific IgE antibodies, a 96-well plate was coated with 50 μL/well antigen solution containing 2.5 μg/mL Biostir AD (Biostir Inc. 20) in 50 mM carbonate–bicarbonate buffer (pH 9.6) overnight at 4 °C. The chemical reagent of the mouse IgE ELISA kit (Mouse IgE ELISA Quantitation Set: Set Bethyl Laboratories, Inc., Montgomery, TX, USA) was used for the enzyme reaction, followed by a labeled antibody reaction according to the manufacturer’s instructions.

### 4.9. Statistical Analysis

Data were analyzed using Microsoft Excel 2019 (MS Excel 2019: Microsoft Corporation, WA, USA). The data were expressed as means ± SD. The unpaired Student’s *t*-test was used to analyze differences between two groups. Values of *p* < 0.05 were considered statistically significant.

## 5. Conclusions

Oral ingestion of unheated yuzu seed oil in NC/Nga AD model mice led to the striking suppression of dermatitis development and elevated serum IgE levels. We suggest that the component involved is limonin, which is present in unheated yuzu seed oil and is below the detection limit in purified yuzu seed oil. Continuous intake of unheated yuzu seed oil, a plant-derived natural product, in healthy adults has been determined as safe after long-term administration [7], underscoring its potential as a viable candidate for treating AD.

## Figures and Tables

**Figure 1 ijms-25-02689-f001:**
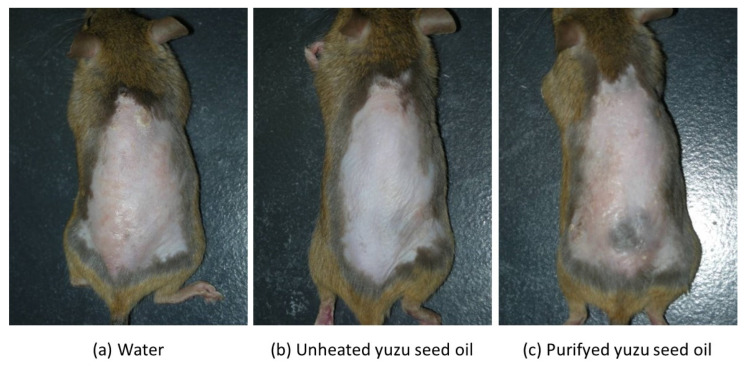
Macroscopic features of *Dermatophagoides farina* body (Dfb)-induced human AD-like skin lesions on the dorsal skin of NC/Nga mice on day 28 following the initiation of sensitization. After completely shaving the dorsal hair and depilation using a hair removal cream, the mice backs were sensitized by applying mite antigen. “Water” is the negative control group. (**a**) The water group (*n* = 5) received 100 µL/mice of water via an oral gastric cannula for rodents once a day for 28 days after sensitization. (**b**,**c**) The unheated and purified yuzu seed oil groups (*n* = 5 each) received 100 µL/mice of unheated and purified yuzu seed oil, respectively, in the same manner.

**Figure 2 ijms-25-02689-f002:**
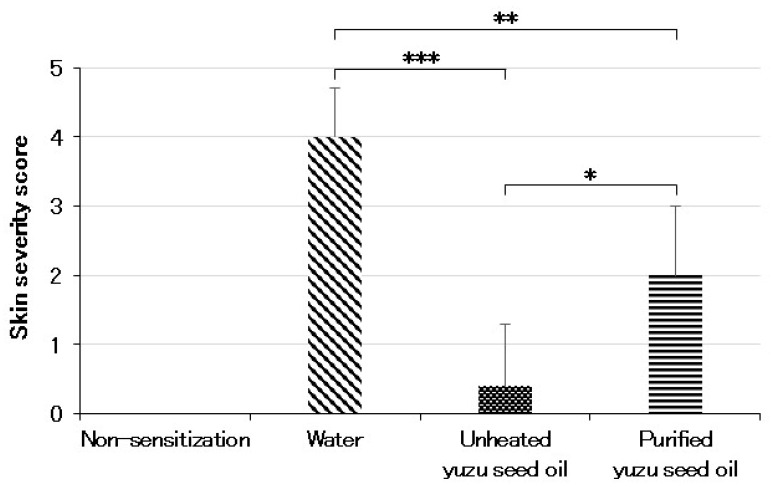
Skin severity score evaluated on day 28 after initiating sensitization by an AD-scoring method. The degree of each symptom (erythema, edema/thickness, hemorrhage/excoriation, and dryness) was scored as 0 (absent), 1 (mild), 2 (moderate), or 3 (severe). The sum of the individual scores was taken as the dermatitis score. In the case of no skin lesion, the score was 0, and the maximum score attainable was 12. Every group represents an average of the scores of five mice. Data are expressed as the means ± SD. The unpaired Student’s *t*-test was used to analyze differences between the two groups. *** *p* < 0.001, ** *p* < 0.01, and * *p* < 0.05. The skin sensitivity score of the non-sensitized group is that of normal mice.

**Figure 3 ijms-25-02689-f003:**
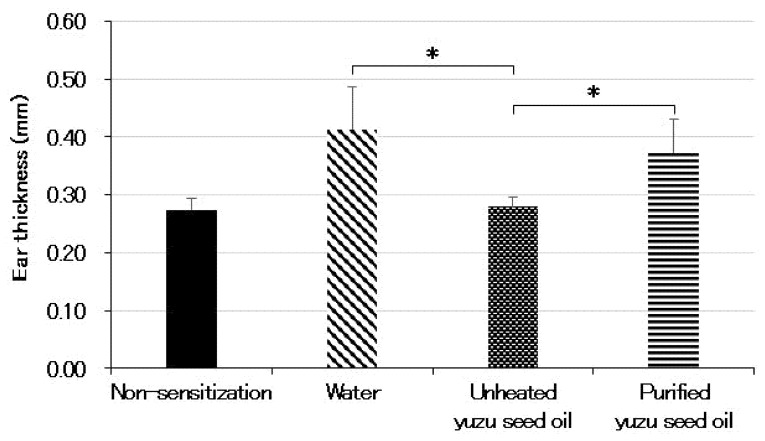
The thickness of the left and right ears were measured five times each with a micrometer equipped with a constant pressure device on day 28 after the initiation of sensitization. The average thickness of both ears was calculated as ear swelling. Every group represents an average of five mice. Data are expressed as the means ± SD. The unpaired Student’s *t*-test was used to analyze differences between the two groups. * *p* < 0.001. The ear thickness of the non-sensitized group is that of normal mouse ears. “Water” is the negative control group. The thickness of the ear indicates the intensity of dermatitis.

**Figure 4 ijms-25-02689-f004:**
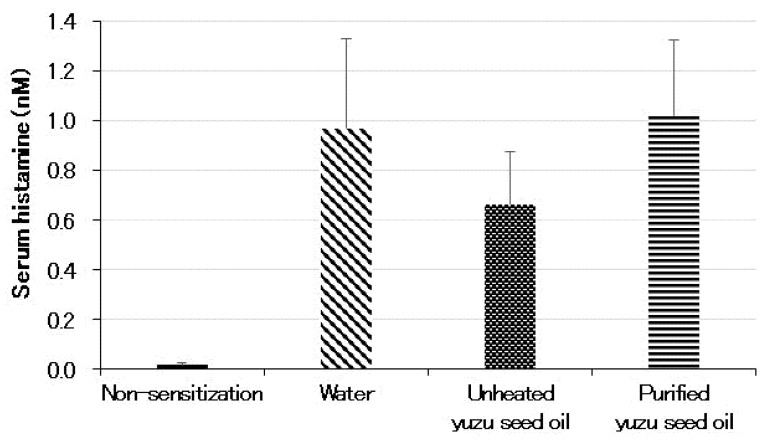
Serum levels of histamine measured using ELISA on day 28 following the initiation of sensitization. Every group represents an average of the data of five mice. Data are expressed as the means ± SD. Histamine levels in the non-sensitized group correspond to those of normal mice.

**Figure 5 ijms-25-02689-f005:**
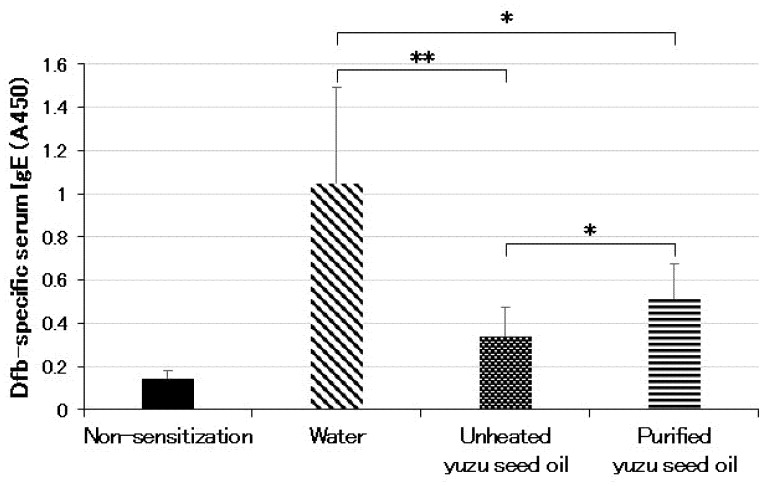
Serum levels of Dfb-specific IgE measured using ELISA on day 28 post the initiation of sensitization. The level of IgE is shown as absorbance at A_450_. Every group represents an average of the serum Dfb-specific IgE level of five mice. Data are expressed as the means ± SD. The unpaired Student’s *t*-test was used to analyze differences between the two groups. ** *p* < 0.01 and * *p* < 0.05. Serum levels of the Dfb-specific IgE in the non-sensitized group correspond to those of normal mice.

**Figure 6 ijms-25-02689-f006:**
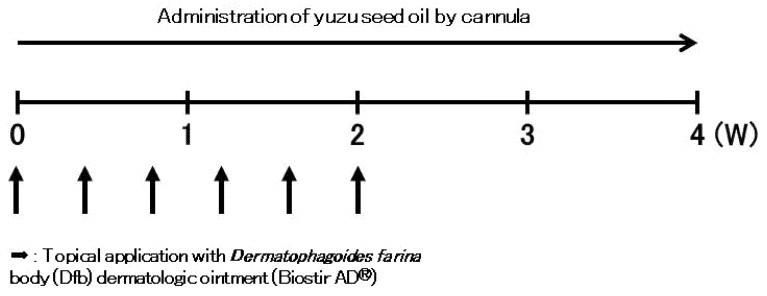
Effects of yuzu seed oil on Dfb-induced AD-like skin lesions in NC/Nga mice. This figure illustrates the experimental schedule of the Dfb-induced human AD-like mouse model. Biostir AD (100 mg) was repeatedly applied to the back and ears three times a week for 2 weeks following the first challenge.

## Data Availability

The data presented in this study are available on request from the corresponding author.

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
