# Peer review of "Oral Ingestion of Yuzu Seed Oil Suppresses the Development of Atopic Dermatitis-like Skin Lesions in NC/Nga Mice"

_ijms, 2024, doi:10.3390/ijms25052689_

Round 1
Reviewer 1 Report
Comments and Suggestions for Authors
The authors of the manuscript have studied the effects of oral ingestion of unheated yuzu seed oil, containing limonin with antioxidant properties, on atopic dermatitis (AD) in an AD mouse model.
Comments
Line 20: specify exact dosage or concentration of yuzu seed oil administered orally to ensure clarity and reproducibility of the results.
Include more details regarding the methodology used for establishing the AD model in NC/Nga mice, mention specific protocol for repeated local exposure to mite antigens. (line 14).
Write size of the clinical trials, as well as any observed adverse effects (line 22).
Write on the extraction process of yuzu seed oil, such as extraction temperature and duration, to enhance reproducibility with relevant references (line 33).
Include information on the concentrations of limonoids in yuzu seed oil and their potential individual contributions to its effects with relevant references (line 36).
Write in brief mechanisms through which unheated yuzu seed oil activates Nrf2 and suppresses ROS production, potentially through specific components, to elucidate its therapeutic pathways (line 47).
Discuss the potential of yuzu seed oil as a natural antioxidant supplement for managing AD, highlighting its promising role in mitigating oxidative stress and inflammation (lines 49-60).
Discuss potential factors contributing to the lack of significant difference in serum histamine levels among the sensitized groups, providing insights into the mechanisms underlying yuzu seed oil's effects on histamine regulation (lines 114-119).
70-79: The significant reduction in clinical skin severity scores and ear thickness in both unheated and purified yuzu seed oil groups indicates a consistent anti-inflammatory effect on AD symptoms, highlighting its therapeutic potential.
In line 168, "Th2 cells release IL-4 and IL-13, leading to IgE class switching in B cells and produc- tion of antigen-specific IgE via the signal transducer and activator of transcription (STAT) pathway [13,35]." - There is a minor formatting issue with the text, with "production" being split into two words. It should be corrected to "production."
Line 222: The fatty acid composition percentages are inconsistent between the two oils. It states 36.8% oleic acid, 35.6% linoleic acid, and 20.1% palmitic acid for unheated yuzu seed oil, but then it lists the same percentages for purified yuzu seed oil in the next sentence. This seems contradictory.
Line 226: The phrase "49.4 mg or more of limonoid nomilin" should be corrected to "49.4 mg or more of limonin".
Line 228: Similar to the previous point, "nomilin, limonin, and obakunon contents were below their detection limits" is stated, but limonin is mentioned as present in the unheated yuzu seed oil. This inconsistency needs clarification or correction.
Line 236: It should be clarified whether the laboratory diet and water were provided ad libitum to the mice.
Line 240: The protocol code "E-00010" might need to be double-checked for accuracy and consistency with the institutional guidelines
Line 292: The reference to "healthy adults" should be corrected to "healthy mice" to maintain consistency with the context of the study, which was conducted on NC/Nga AD model mice, not humans.
A plagiarism check is recommended
Comments on the Quality of English LanguageModerate editing of English language required
Reviewer 2 Report
Comments and Suggestions for Authors
Journal IJMS (ISSN 1422-0067)
Manuscript: ID ijms-2872947
Type: Article
Title: Oral ingestion of yuzu seed oil suppresses the development of atopic dermatitis-like skin lesions in NC/Nga mice
Authors: Kimito Asano , Yoshiya Watanabe , Mio Miyamoto , Mochifumi Toutani , Shunji Mizobuchi *
Section: Molecular Pathology, Diagnostics, and Therapeutics
As atopic dermatitis is a debilitating chronic inflammatory skin disease that affects especially children, great efforts are being made to identify efficient therapies with as few adverse events as possible.
This research explores the antioxidant and anti-inflammatory effects of limonoids contained in yuzu seed oil, suggesting its potential use in patients with AD. It has previously been shown that long-term oral intake of unheated yuzu seed oil in humans is safe and it reduces lipid peroxides in the blood.
The originality of the research consists of exploring the efficacy of unheated yuzu seed oil compared with purified seed oil and water. The results showed that clinical improvement was more significant in the unheated yuzu seed oil group.
In the introduction, lines 21-22 please reformulate: “Oral ingestion of yuzu seed oil in NC/Nga AD model mice significantly suppressed dermatitis deterioration and increased serum IgE levels”. You wanted to say that it decreases the IgE levels.
Line 31. Space “upto” 9% of the fruit weight
Line 51: “itchy eczema” is a pleonasm. Eczema is per se itchy.
Conclusions are consistent with the data presented. References are appropriate.
Round 2
Reviewer 1 Report
Comments and Suggestions for Authors
No further comment